# Trends of maternal waterpipe, cigarettes, and dual tobacco smoking in Jordan. A decade of lost opportunities

**Khalid A. Kheirallah**●*, **Nuha Shugaa Addin, Maan M. Alolimat**

Department of Public Health, Faculty of Medicine, Jordan University of Science and Technology, Irbid, Jordan

* kkheiral@gmail.com

## Abstract

### Background

Maternal tobacco use is a global public health problem. In the literature, the focus was mainly on cigarette smoking, minimally on waterpipe use, and totally ignored dual use among pregnant women. We estimated the prevalence of current maternal tobacco use by tobacco product (cigarette, waterpipe, and dual use) over a period of ten years (2007 to 2017), and examined the socio-demographic patterning of maternal tobacco use.

### Methods

A secondary analysis of Jordan DHS four data waves was conducted for women who reported to be pregnant at the time of the survey. Current cigarette and waterpipe tobacco use were investigated. Prevalence estimates for cigarette-only, waterpipe-only, and dual use, as well as for cigarette, regardless of waterpipe, and waterpipe, regardless of cigarette, were reported. The effect of independent variables on cigarette smoking, waterpipe use, and dual use was assessed. Logistic regression models assessed the adjusted effects of socio-demographic variables on cigarette smoking, waterpipe use, and on dual use. For each outcome variable, a time-adjusted and a time-unadjusted logistic models were conducted.

### Results

Over the last decade, the prevalence estimates of current cigarette-only smoking slightly decreased. The prevalence estimates of current waterpipe-only use exceeded those for cigarette-only after 2007 and showed a steady overall increase. Current dual use showed a continuous rise especially after 2009. Gradual increase in cigarette smoking (4.1%, in 2007, and 5.7% in 2017) and in waterpipe use (2.5% to 6.4%) were detected. Education showed an inverse relationship with cigarette and waterpipe smoking. Household wealth demonstrated a positive association with cigarette and waterpipe smoking.

**Data Availability Statement:** Data is publicly available online at: https://dhsprogram.com/data/available-datasets.cfm. Authors and readers can create an account and download the datasets

associated with the study. No special access privileges were granted to the authors.

**Funding:** The authors received no specific funding for this work.

**Competing interests:** The authors have declared that no competing interests exist.

## Conclusions

Tobacco use epidemic is expanding its roots among pregnant women in Jordan through not only waterpipe use but also dual cigarette–waterpipe smoking. Maternal and child services should consider tobacco counseling and cessation.

## Introduction

Tobacco use (cigarette or waterpipe) is on the rise throughout the world despite continuous interventional programs and anti-tobacco campaigns [1]. It is deemed to be a global public health problem that attributes to numerous morbidities and mortalities [2–5]. During pregnancy, tobacco use is a public health priority as pregnant women represent a highly vulnerable population. Tobacco use during pregnancy is associated with a wide variety of maternal as well as neonatal adverse effects. Several studies have shown that cigarette smoking during pregnancy is linked to higher rates of spontaneous abortion [6], ectopic pregnancy, premature rupture of membranes [7], placenta abruption [8], and placenta previa [9]. In addition, babies born to women who smoke cigarettes are more prone to being preterm and of low birth weight (less than 2500 g) leading to increased perinatal morbidity and mortality [10]. Maternal waterpipe tobacco use has been associated with decreasing newborn's body weight as well as anthropometric measurements [11, 12]. Evidence also suggests that maternal cigarette smoking during pregnancy can affect the postnatal period by increasing the risk of infant respiratory illnesses resulting in sudden unexpected infant death (SUID) [13]. Interestingly, the complications can extend beyond pregnancy and neonatal period to cause long term health effects such as childhood obesity and metabolic disorders [14], asthma [15], dental caries [16], attention deficit hyperactivity disorder (ADHD) [17] and decreased lung function in adolescents [18].

In the Arab region, countries like Jordan, Syria, Lebanon, and Palestine are taking the lead in cigarette smoking prevalence estimates. Among women, Lebanon takes the lead followed by Jordan then Syria [19]. Beside cigarette smoking, waterpipe tobacco use is becoming more popular and the Arabian youth numbers of its users are increasing among girls in specific, especially in Jordan, Lebanon, and the West Bank [20]. Notably, the trend of tobacco use among women in Jordan showed a steady cigarette smoking pattern and a steady increase in waterpipe use between 2002 and 2012 [21]. This increase in waterpipe prevalence has been suggested to be due to the increasing social acceptability and the misconception that waterpipe use is less harmful compared to cigarette smoking [22, 23].

Pregnancy is considered an incentive time known as "teachable moment" that motivates women to modify their behavior and quit unhealthy habits including tobacco use [24]. However, many pregnant women continue to smoke. Studies have shown variations in the prevalence of tobacco use during pregnancy regionally and across different countries. A systematic review and meta-analysis aiming to estimate the prevalence of cigarette smoking during pregnancy reported that the global prevalence was estimated at 1.7%, being highest in the European region with 8.1%. The prevalence in the Eastern Mediterranean region was estimated at 0.9%. Ireland had the highest prevalence worldwide with an estimate of 38.4% [25]. A national sample of pregnant women in the USA showed that the prevalence of cigarette smoking among pregnant women was 13.8% [26]. In Lebanon, the prevalence of tobacco use among pregnant women was estimated at 17% for cigarettes-only smoking, 4% for waterpipe-only use, and 1.5% for dual (cigarettes and waterpipe) smoking [27]. In Jordan, a cross-sectional study that randomly selected pregnant women from maternity clinics in middle and north the kingdom

showed that 7.9% of women were current cigarette smokers and 8.7% were current waterpipe smokers [28].

In order to develop tailored smoking cessation programs aiming to encourage women to quit tobacco use during pregnancy, there is a need to estimate the national prevalence of tobacco use by product and also to identify factors associated with its use. There have been several studies investigating the predictors of tobacco use among pregnant women including socio-demographic, relationship-related, smoking-related, as well as psychological-related factors [29–31]. With regard to socio-demographic characteristics, tobacco use during pregnancy has been associated with age, education, ethnicity, income and marital status [32]. A nationally representative cohort of women demonstrated an adverse association between cigarette smoking during pregnancy and maternal education and income level [33]. With respect to age, there are inconsistencies between different studies, some reported an association between older maternal age and cigarette smoking cessation [34, 35]. Others reported older age to be associated with a higher likelihood of cigarette smoking during pregnancy [36, 37].

Although the adverse health effects of smoking during pregnancy are well established, to the best of our knowledge, no national estimates exist of the prevalence of tobacco use by product (cigarette, waterpipe, and dual use) during pregnancy in Jordan, or the region. Examining the pattern of maternal cigarette smoking, waterpipe smoking, as well as dual-use, and how these patterns change over time, is of critical significance, especially in developing countries. Additionally, cigarette smoking was mainly addressed in the literature with limited studies focusing on waterpipe or dual use. The identification of the predictors of maternal tobacco use is very crucial in order to determine the groups of women that are at higher risk of tobacco use to develop data-driven appropriate interventions and establish evidence-based national policies to combat tobacco use among pregnant women.

Utilizing four waves of the Jordan Demographic and Health Surveys (DHS 2007, 2009, 2012, and 2017), the current study aims to: estimate the prevalence of current maternal tobacco use by tobacco product (cigarette, waterpipe, and dual use) and socio-demographic patterns, and to investigate trends in these patterns over a period of ten years (2007 to 2017).

## Materials and methods

### Sample and data

We utilized a subset of data from each of the four waves of Jordan DHS for the years 2007, 2009, 2012, and 2017. Jordan DHS provides a nationally representative sample, utilizing the Jordan's Population and Housing Census (JPHC), by including urban and rural areas of all twelve governorates which are grouped into north, central, and south regions. All DHS data waves used a multi-stage stratified sampling technique proportional to size. This sampling design aimed to produce a nationally representative sample for all households in Jordan. The first sampling stage is geographically based and was proportional to the cluster size. Each governorate was divided into smaller administrative units: districts, sub-districts, localities, areas, and sub-areas. Each sub-area was then divided into census blocks. Data from each block regarding households, populations, geographical locations, and socio-demographic characteristics was already available from the JPHC. The census blocks are regrouped to form a general statistical unit of moderate size, called a cluster, which is widely used in various surveys as the primary sampling unit (PSU). In the first stage, clusters were selected with probability proportional to cluster size (the number of residential households enumerated in the JPHC). After that, a household listing within each PSU was carried out. The resulting household lists served as the sampling frame for selecting households in the second stage. A fixed number of households per cluster was selected with an equal probability systematic selection from the newly

created household listing. Within each selected household, all ever-married women (age 15–49), who were either residents of the selected households or visitors who stayed in the households the night before the survey, were eligible for interview. The main questionnaires used in the Jordan DHS are the Household Questionnaire, the Woman's Questionnaire, the Man's Questionnaire, and the Biomarker Questionnaire (used only in 2017 data). These standardized English questionnaires are based on the DHS Program and were minimally adjusted to reflect issues relevant to Jordan [38]. The questions were translated into Arabic and tested to ensure clarity. Interviews were administered by well-trained female workers.

In the present study, only one questionnaire from each wave was used, the women's questionnaire, which included ever-married women. Since we were interested in the prevalence of tobacco use among pregnant women, we only included pregnant women in our analyses. Additionally, we excluded women between the ages of 45–49 years because there was no pregnancy above that age in 2009 and 2012 and only 0.1% and 0.2% in 2007 and 2017, respectively. Thus, pregnant women comprised 13.6%, 12.9%, 11.1%, and 12.2% of the total women in the 2007, 2009, 2012, and 2017 Jordan DHS datasets, respectively, from ever married women sample aged 15–44 years (9,637 in 2007, 8,835 in 2009, 9,782 in 2012, and 12,269 in 2017). The women questionnaire included several topics, namely background characteristics, maternal and child health, and behaviors related to certain topics like tobacco use (cigarette smoking and waterpipe, Nargila, use). We focused on tobacco use among pregnant women and its relation to socio-demographic characteristics.

## Measures

Our outcome measure among pregnant women at the time of the survey is current tobacco use (cigarette, waterpipe, and dual use). We calculated the outcomes from the two dichotomous (Yes or No) questions: "Do you *currently* smoke cigarette?" and "Do you *currently* smoke Nargila (waterpipe)?". The independent variables were socio-demographic determinants including age group, residency, education, and household wealth index. Age was coded into three categories: 15–24, 24–34, and 35–44 years. The residency was coded as rural and urban. Education was coded into three categories: primary or less, secondary, and higher than secondary. As for the household wealth index, we used the same quintile classification of the DHS data with the following tags: Poorest, poor, middle, rich, and richest. The wealth index was calculated by using observational data on ownership of household assets, goods, and services.

## Statistical analysis

We performed the statistical analyses using IBM Statistical Package for Social Sciences (SPSS) version 21. Because of non-proportional number of the study population across the waves, we used sampling weights to acquire representative percentages for the complex multi-stage design of the DHS. Sample weights are already provided in each dataset. At the beginning, we calculated the prevalence of pregnant women aged 15–44 years. Then we estimated the prevalence of tobacco smoking by product (cigarette-only, waterpipe-only, and dual use as well as cigarettes, regardless of waterpipe, and waterpipe, regardless of cigarettes) among pregnant women in each of the four waves. We presented the prevalence estimates of tobacco use by socio-demographic variables across the four waves. In the logistic regression models, we used three main binary outcome variables: current cigarette smoking, current waterpipe use, and dual use (yes *vs* no for each variable). We pooled data from all four waves and assessed the association between socio-demographic characteristics and tobacco use variables using a set of logistic regression models. Results were reported as adjusted odds ratios (AORs) and 95% confidence intervals (C.I.). A p-value of 0.05 was considered as statistically significant in all cases.

In order to test interaction between time (year, as a categorical variable) and socio-demographic measures, we conducted both time-unadjusted and time-adjusted logistic models for each outcome measure. We also tested for multicollinearity between independent variables controlled for in the analysis and the assumption of reasonable independence was met since the variance inflation factors (VIF) between variables were less than five [39].

## Results

### Socio-demographic characteristics

This study included 13.6%, 12.9%, 11.1%, and 12.2% of the total women interviewed in the Jordan DHS samples for 2007, 2009, 2012, and 2017, respectively (Table 1). As shown in Table 2, a total of 1,313 pregnant women between 15 and 44 years old in 2007, 1,137 in 2009, 1,084 in 2012, and 1,493 in 2017 were included in the current analyses. Socio-demographic characteristics of pregnant women in the four waves of DHS data showed that the mean age (standard deviation) of pregnancy was 28.5 (6.1) years in 2007, 27.9 (5.9) in 2009, 27.6 (6.0) in 2012, and 27.6 (6.0) in 2017. In all waves, around 50% of participants were between the ages of 25 and 34 years. The majority of participants resided in urban areas (83.8% in 2007, 80.9% in 2009, 82.8% in 2012, and 89.4% in 2017). The sampling distribution by education showed that, in each wave, more than half of the study participants had a secondary education level and at least 30.7% had higher education; only a small proportion of women had primary education or less. With respect to household wealth status (index), in all waves, a range between 18.2% and 25.9%, 20.5% and 27.5%, 17.4% and 26.4%, 17.3% and 20.0%, and 11.0% and 16.6% of pregnant women were in the poorest, poor, middle, rich, and the richest households, respectively.

### Tobacco use prevalence estimates

Table 3 shows the prevalence estimates of current maternal tobacco use in Jordan for the last decade. Across the four waves, the prevalence estimates for current cigarette-only smoking did not change much between 2007 and 2017 except for a slight decrease in 2009 (3.3% in 2007, 2.0% in 2009, 2.5% in 2012, and 2.9% in 2017). The prevalence estimates of current waterpipe-only use, on the other hand, exceeded the prevalence of current cigarette-only smoking except in 2007. Current waterpipe-only use prevalence estimate showed a steady increase from 2007 to 2012, where it became double the estimate for current cigarette-only smoking among study participants, before decreasing notably in 2017 (1.7% in 2007, 3.2% in 2009, 5.1% in 2012, and 3.6% in 2017) (Fig 1). Interestingly, the prevalence estimate of current dual use has showed a continuous rise across the four waves being 0.8%, 1.0%, 2.0%, and 2.8% in 2007, 2009, 2012, and 2017, respectively.

**Table 1. Prevalence of pregnant women in Jordan across the four waves of DHS data.**

| DHS Wave (Year) | Pregnancy Status | | |
|---|---|---|---|
| | N (%) | | |
| | Non-pregnant | Pregnant | Total |
| Wave 1 (2007) | 8,324 | 1,313 | 9,637 |
| | *86.4%* | *13.6%* | *100%* |
| Wave 2 (2009) | 7,699 | 1,136 | 8,835 |
| | *87.1%* | *12.9%* | *100%* |
| Wave 3 (2012) | 7,697 | 1,085 | 9,782 |
| | 88.9% | 11.1% | 100% |
| Wave 4 (2017) | 10,776 | 1,493 | 12,269 |
| | *87.9%* | *12.2%* | *100%* |

**Table 2. Socio-demographics of pregnant women across the four waves of DHS data.**

| | 2007 | 2009 | 2012 | 2017 |
|---|---|---|---|---|
| | N = 1,313 | N = 1,137 | N = 1,084 | N = 1,493 |
| | N (%) | N (%) | N (%) | N (%) |
| **Age** | | | | |
| 15–24 | 376 (28.6) | 361 (31.8) | 396 (36.5) | 514 (34.4) |
| 25–34 | 692 (52.7) | 624 (52.9) | 517 (47.6) | 771 (51.6) |
| 35–44 | 245 (18.7) | 152 (13.4) | 172 (15.9) | 208 (13.9) |
| **Residence** | | | | |
| Urban | 1101 (83.8) | 919 (80.9) | 898 (82.8) | 1334 (89.4) |
| Rural | 213 (16.2) | 217 (19.1) | 186 (17.2) | 159 (10.6) |
| **Education** | | | | |
| < Primary | 93 (7.1) | 89 (7.8) | 72 (6.6) | 114 (7.6) |
| Secondary | 817 (62.2) | 655 (57.7) | 611 (56.3) | 751 (50.3) |
| Higher | 403 (30.7) | 392 (34.5) | 402 (37.1) | 628 (42.1) |
| **Household wealth** | | | | |
| Poorest | 332 (25.3) | 270 (23.8) | 197 (18.2) | 387 (25.9) |
| Poor | 361 (27.5) | 259 (22.8) | 222 (20.5) | 327 (21.9) |
| Middle | 228 (17.4) | 286 (25.2) | 286 (26.4) | 267 (18.5) |
| Rich | 235 (17.9) | 196 (17.3) | 199 (18.4) | 298 (20.0) |
| Richest | 157 (12.0) | 125 (11.0) | 180 (16.6) | 104 (13.7) |

Prevalence estimates for current cigarette smoking, regardless of waterpipe use, ranged between 4.1%, in 2007, and 5.7% in 2017. These estimates showed gradual increase across the study waves except for 2009 (3.0%). Similarly, gradual increase in the prevalence estimates of current waterpipe use, regardless of cigarette smoking, was noted: from 2.5%, in 2007, to 6.4% in 2017. In 2012, the highest estimate was reported for waterpipe use (7.1%). Current tobacco use estimates, regardless of type, showed gradual increase from 5.8%, in 2007, to 9.3% in 2017 with the highest estimates reported in 2012 (9.6%) (Fig 2).

## Prevalence of tobacco use by socio-demographic characteristics

Table 4 shows differences in the prevalence estimates of current tobacco use among pregnant women by socio-demographic characteristics. Tobacco use predominantly increased among the age group 25–34 years in 2007 and 2009. In 2012, maternal cigarette smokers were

**Table 3. Prevalence of tobacco use among pregnant women across the four waves of Jordan DHS data.**

| DHS Wave (Year) | Tobacco use (n, %) | | | | | | | |
|---|---|---|---|---|---|---|---|---|
| | None | Cigarettes-only | WP-only | Dual use | Total | Cigarettes | WP | Tobacco use (cigarette or waterpipe) |
| **Wave 1** | **1,237** | **43** | **22** | **11** | **1,313** | **54** | **33** | **76** |
| *(2007)* | *94.2%* | *3.3%* | *1.7%* | *0.8%* | *100.0%* | *4.1%* | *2.5%* | *5.8%* |
| **Wave 2** | **1,067** | **23** | **36** | **11** | **1,137** | **34** | **47** | **70** |
| *(2009)* | *93.8%* | *2.0%* | *3.2%* | *1.0%* | *100.0%* | *3.0%* | *4.2%* | *6.2%* |
| **Wave 3** | **980** | **27** | **55** | **22** | **1,084** | **49** | **77** | **104** |
| *(2012)* | *90.4%* | *2.5%* | *5.1%* | *2.0%* | *100.0%* | *4.5%* | *7.1%* | *9.6%* |
| **Wave 4** | **1,354** | **43** | **54** | **42** | **1,493** | **85** | **96** | **139** |
| *(2017)* | *90.7%* | *2.9%* | *3.6%* | *2.8%* | *100.0%* | *5.7%* | *6.4%* | *9.3%* |

WP: Waterpipe

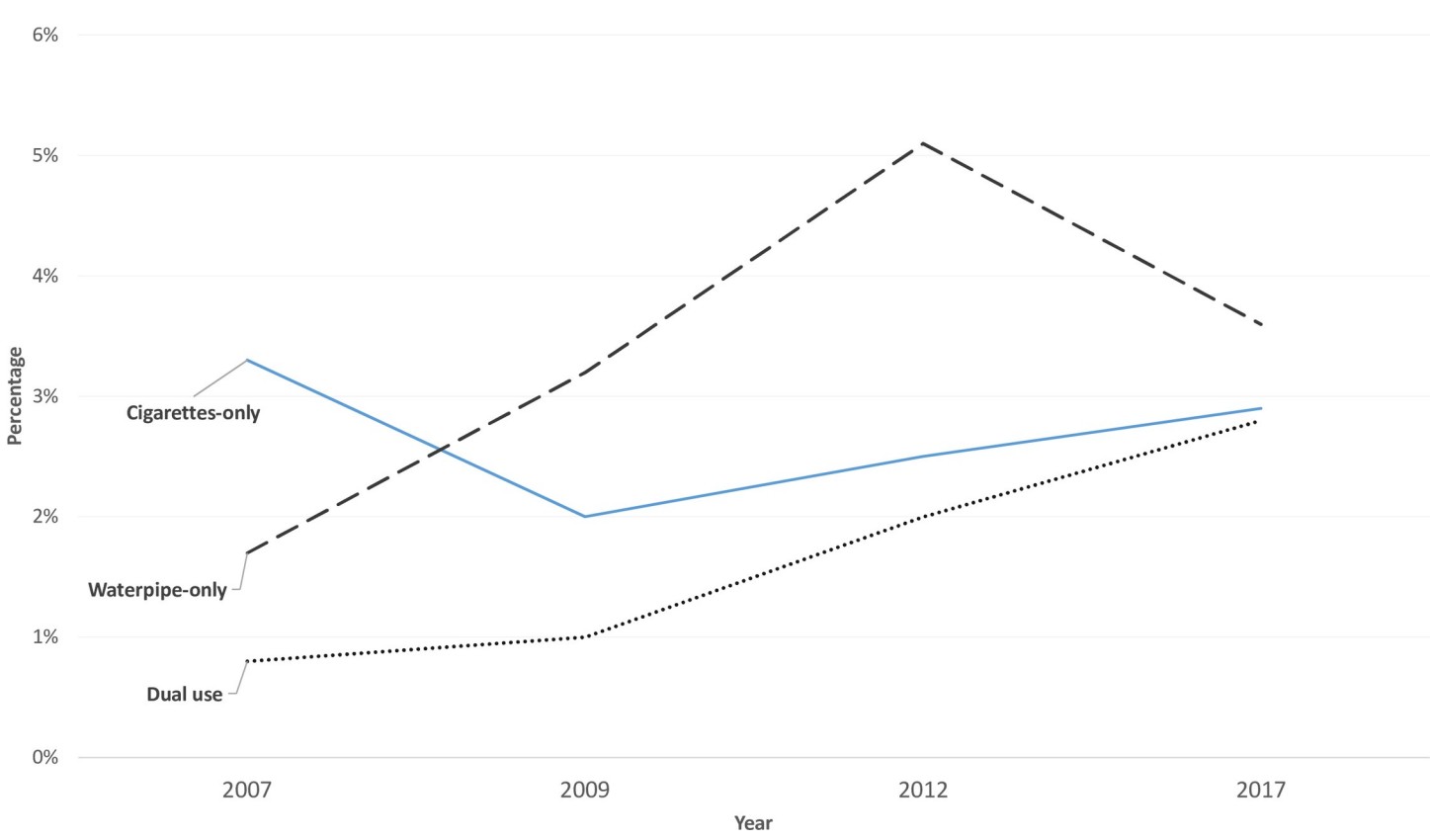

**Fig 1. Prevalence estimates of cigarette-only, waterpipe-only, and dual use among ever-married pregnant women between 2007 and 2017.**

significantly higher among the age group 25–34 years while maternal waterpipe users among 15-24-year age group. Current tobacco use estimates did not vary with age in 2017 and with residency in all waves of DHS.

Current cigarette smoking displayed a significant inverse association with education, being higher among primary educated pregnant women in all four waves. There was a gradual decrease in cigarette smoking as education increased; prevalence estimate decreased from 10.8% for primary education or less to 1.7% in higher education group in 2007; from 6.7% to 0.5% in 2009; from 5.6% to 1.7% in 2012; and from 6.1% to 1.6% in 2017. Current waterpipe use also showed an inverse association with education but only in 2009 and 2012; prevalence estimate decreased from 5.6% for less primary education to 3.3% in higher education group in 2009 and from 8.3% to 5.0% in 2012. In 2007 and 2017, current waterpipe use showed a positive association with education; prevalence was higher among the higher and secondary educated pregnant women, respectively.

Current cigarette smoking showed a significantly inverse relationship with household wealth in all four waves except 2009. There was an increase in prevalence estimate of cigarette smoking with poor household wealth (5.5%, 3.1%, and 4.3% in 2007, 2012, and 2017, respectively). In 2009, however, cigarette smoking prevalence estimate was significantly higher among pregnant women living in rich households (7.1%). Conversely, current waterpipe use estimate displayed a positive association with household wealth; prevalence estimate significantly increased among the richest with 6.4% in 2007; 6.5% in 2009; 9.4% in 2012. In 2017, current waterpipe use shows higher prevalence estimate among both poor and rich pregnant smokers (5.2% and 5.1%, respectively).

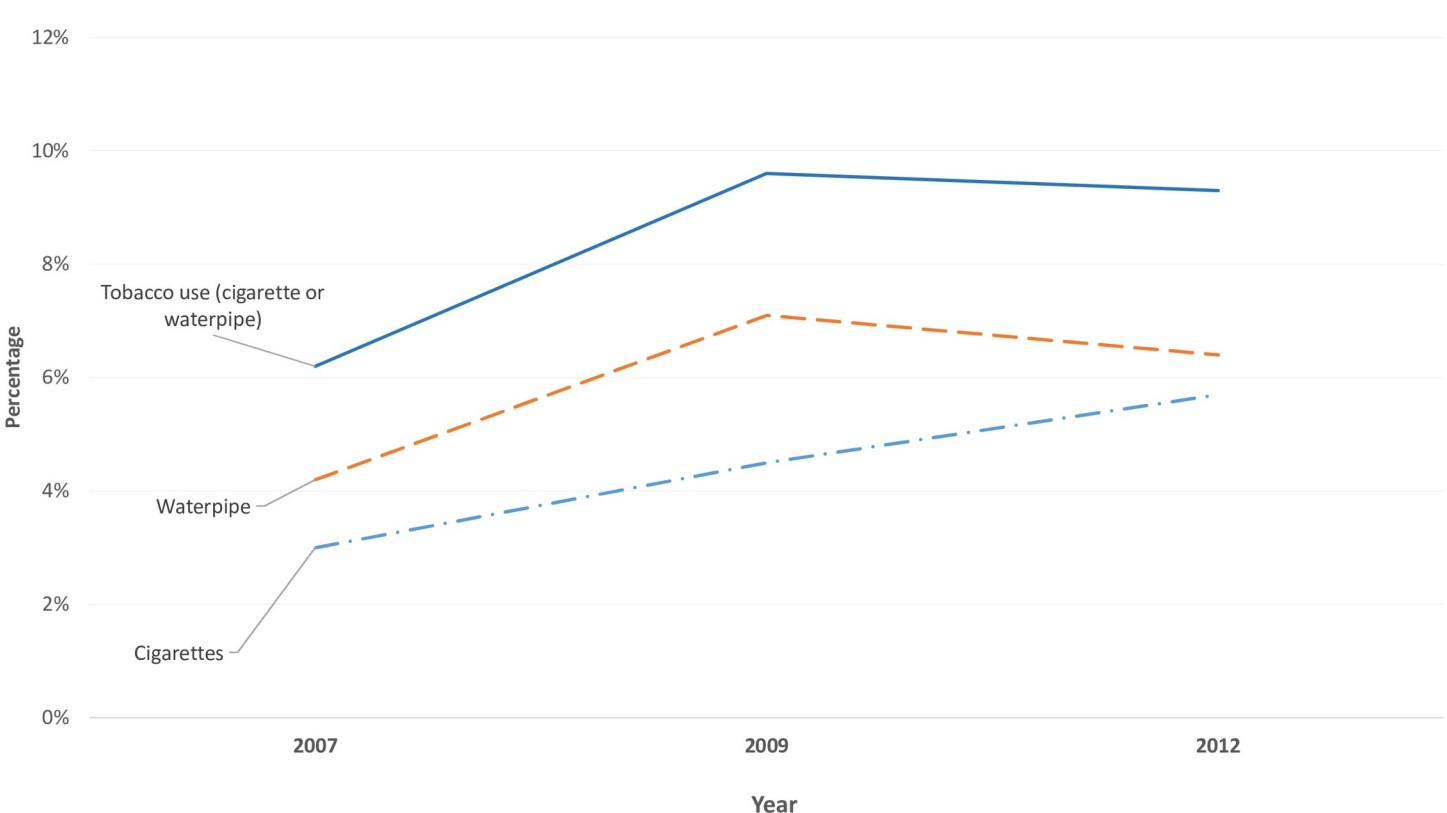

**Fig 2. Prevalence estimates of cigarette smoking, waterpipe use, and tobacco use (regardless of type) among ever-married pregnant women between 2007 and 2017.**

### Predictors of current cigarette, waterpipe, and dual smoking

Table 5 presents the results of pooled logistic regression models conducted to estimate the odds of current maternal tobacco use by socio-demographic characteristics. The time-unadjusted (1) and time-adjusted (2) models demonstrate near-identical odds ratio.

Age group 25–34 years predicted higher odds of current cigarette and dual smoking. Conversely, increasing age demonstrated a protective effect against current waterpipe use. Living in rural areas predicted lower odds of current cigarette and waterpipe smoking compared to the urban residence. Increasing education significantly predicted lower odds of current cigarette smoking. Pregnant women with secondary education and higher were less likely to be current cigarettes smokers compared to their counterparts in the primary education or less category. Similarly, pregnant women with higher education were significantly less likely to be current waterpipe users in comparison to primary education or less category.

Household wealth displayed a positive association with current cigarette and waterpipe smoking. Pregnant women in the middle and rich category were more likely to be current cigarettes smokers compared to the poorest category. Similarly, pregnant women in the middle, rich, and richest categories predicted significantly higher odds of current waterpipe use compared to the poorest category.

Finally, time exhibited a limited effect of cigarette smoking as the odds of current cigarette smoking were statistically significantly only in 2017, compared to 2007. Still, time exhibited a significant effect of current waterpipe and dual smoking. The odds of current waterpipe use and dual smoking increased between 2007 and 2017. In particular, the odds of waterpipe use and dual smoking were 2.59 and 3.42 times, respectively, higher in 2017 compared to 2007.

**Table 4. Tobacco use by socio-demographics among pregnant women across the four waves of DHS data.**

| | 2007 | | | | | 2009 | | | | | 2012 | | | | | 2017 | | | | |
|---|---|---|---|---|---|---|---|---|---|---|---|---|---|---|---|---|---|---|---|---|
| | N = 1313 | | | | | N = 1,136 | | | | | N = 1,085 | | | | | N = 1,493 | | | | |
| | Tobacco use* n (%) | | | | | Tobacco use* n (%) | | | | | Tobacco use* n (%) | | | | | Tobacco use* n (%) | | | | |
| | None | Cig. only | WP only | Dual | P-value | None | Cig. only | WP only | Dual | P-value | None | Cig. only | WP only | Dual | P-value | None | Cig. only | WP only | Dual | P-value |
| **Overall estimates** | 1237 (94.2) | 43 (3.3) | 22 (1.7) | 11 (0.8) | <0.001 | 1067 (93.8) | 23 (2.0) | 36 (3.2) | 11 (1.0) | <0.001 | 980 (90.4) | 27 (2.5) | 55 (5.1) | 22 (2.0) | <0.001 | 1354 (90.7) | 43 (2.9) | 54 (3.6) | 42 (2.8) | <0.001 |
| **Age** | | | | | | | | | | | | | | | | | | | | |
| 15–24 | 363 (96.5) | 7 (1.9) | 6 (1.6) | 0 (0.0) | 0.003 | 343 (95.0) | 2 (0.6) | 11 (3.0) | 5 (1.4) | 0.029 | 350 (88.4) | 6 (1.5) | 32 (8.1) | 8 (2.0) | 0.001 | 474 (92.2) | 11 (2.1) | 20 (3.9) | 9 (1.8) | 0.156 |
| 25–34 | 635 (91.9) | 32 (4.6) | 16 (2.3) | 10 (1.4) | | 575 (92.1) | 19 (3.0) | 24 (3.8) | 6 (1.0) | | 466 (90.0) | 16 (3.1) | 21 (4.1) | 15 (2.9) | | 691 (89.7) | 29 (3.8) | 28 (3.6) | 22 (2.9) | |
| 35–44 | 239 (97.6) | 4 (1.6) | 1 (0.4) | 1 (0.4) | | 149 (98.0) | 2 (1.3) | 1 (0.7) | 0 (0.0) | | 164 (95.9) | 5 (2.9) | 2 (1.2) | 0 (0.0) | | 189 (90.4) | 4 (1.9) | 6 (2.9) | 10 (4.8) | |
| **Residence** | | | | | | | | | | | | | | | | | | | | |
| Urban | 1034 (93.9) | 35 (3.2) | 22 (2.0) | 10 (0.9) | 0.390 | 858 (93.3) | 17 (1.8) | 34 (3.7) | 11 (1.2) | 0.05 | 804 (89.4) | 24 (2.7) | 50 (5.6) | 21 (2.3) | 0.19 | 1203 (90.1) | 42 (3.1) | 50 (3.7) | 40 (3.0) | 0.240 |
| Rural | 203 (95.3) | 8 (3.8) | 1 (0.5) | 1 (0.5) | | 209 (96.3) | 6 (2.8) | 2 (0.9) | 0 (.00) | | 176 (94.6) | 3 (1.6) | 5 (2.70) | 2 (1.1) | | 151 (95.0) | 2 (1.3) | 4 (2.5) | 2 (1.3) | |
| **Education** | | | | | | | | | | | | | | | | | | | | |
| Primary or less | 81 (87.1) | 10 (10.8) | 0 (0.0) | 2 (2.2) | <0.001 | 79 (87.8) | 6 (6.7) | 5 (5.6) | 0 (0.0) | 0.013 | 62 (86.1) | 4 (5.6) | 6 (8.3) | 0 (0.0) | 0.008 | 102 (89.5) | 7 (6.1) | 3 (2.6) | 2 (1.8) | 0.018 |
| Secondary | 773 (94.6) | 26 (3.2) | 10 (1.2) | 8 (1.0) | | 616 (94.0) | 15 (2.3) | 19 (2.9) | 5 (0.8) | | 543 (89.0) | 17 (2.8) | 29 (4.8) | 21 (3.4) | | 696 (89.1) | 27 (3.6) | 33 (4.4) | 22 (2.9) | |
| Higher | 383 (95.0) | 7 (1.7) | 12 (3.0) | 1 (0.2) | | 372 (94.7) | 2 (0.5) | 13 (3.3) | 6 (1.5) | | 375 (93.1) | 7 (1.7) | 20 (5.0) | 1 (0.2) | | 583 (92.8) | 10 (1.6) | 18 (2.9) | 17 (2.7) | |
| **Wealth index** | | | | | | | | | | | | | | | | | | | | |
| Poorest | 316 (95.2) | 10 (3.0) | 0 (0.0) | 6 (1.8) | <0.001 | 259 (95.6) | 3 (1.1) | 5 (1.8) | 4 (1.5) | <0.001 | 185 (93.4) | 6 (3.0) | 2 (1.0) | 5 (2.5) | <0.001 | 365 (94.6) | 10 (2.6) | 8 (2.1) | 3 (0.8) | 0.002 |
| Poor | 338 (93.4) | 20 (5.5) | 3 (0.8) | 1 (0.3) | | 247 (95.7) | 3 (1.2) | 8 (3.1) | 0 (0.0) | | 213 (95.5) | 7 (3.1) | 2 (0.9) | 1 (0.4) | | 289 (88.1) | 14 (4.3) | 17 (5.2) | 8 (2.4) | |
| Middle | 210 (92.5) | 6 (2.6) | 7 (3.1) | 4 (1.8) | | 267 (93.4) | 3 (1.0) | 10 (3.5) | 6 (1.2) | | 248 (86.7) | 9 (3.1) | 21 (7.3) | 8 (2.8) | | 255 (92.1) | 9 (3.2) | 4 (1.4) | 9 (3.2) | |
| Rich | 231 (98.7) | 1 (0.4) | 2 (0.9) | 0 (0.0) | | 177 (90.3) | 14 (7.1) | 5 (2.6) | 0 (0.0) | | 174 (86.0) | 6 (3.0) | 13 (6.5) | 7 (3.5) | | 267 (89.9) | 6 (2.0) | 15 (5.1) | 9 (3.0) | |
| Richest | 141 (89.8) | 5 (3.2) | 10 (6.4) | 1 (1.6) | | 116 (93.5) | 0 (0.0) | 8 (6.5) | 0 (0.0) | | 161 (89.4) | 1 (0.6) | 17 (9.4) | 1 (0.6) | | 178 (86.8) | 4 (2.0) | 10 (4.9) | 13 (6.3) | |

* Cig. only: Cigarette-only, WP only: Waterpipe-only, Dual: Cigarettes and waterpipe.

## Discussion

To our knowledge, this study is the first to provide estimates of current poly tobacco use (cigarette, waterpipe, and dual) among pregnancy using nationally representative samples. The study, therefore, sheds light on a critical public health problem, in Jordan and the region, by assessing the long-term trends of the maternal tobacco use by product and its relation to the socio-demographic determinants. The results report alarming tobacco use estimates (especially for waterpipe use and dual smoking) among pregnant women and suggest that while current tobacco use prevalence estimates almost doubled, between 2007 and 2017, current waterpipe and dual use estimates increased by about three folds. This may indicate a surge of maternal waterpipe use especially in the middle age group. Further, estimates of waterpipe use exceeded those for cigarette smoking since 2009. Dual use (cigarette and waterpipe), on the other hand,

**Table 5. Time-unadjusted (Model 1) and time-adjusted (Model 2) predictors of cigarette, waterpipe, and dual smoking among pregnant women in Jordan.**

| | | Cigarette smoking | | | | Waterpipe use | | | | Dual smoking | | | |
| --- | --- | --- | --- | --- | --- | --- | --- | --- | --- | --- | --- | --- | --- |
| | | Model 1 | | Model 2 | | Model 1 | | Model 2 | | Model 1 | | Model 2 | |
| | | AOR* | C.I.^ | AOR* | C.I.^ | AOR* | C.I.^ | AOR* | C.I.^ | AOR* | C.I.^ | AOR* | C.I.^ |
| **Age (years)** | | | | | | | | | | | | | |
| 15–24 | | 1.00 | Ref | 1.00 | Ref | 1.00 | Ref | 1.00 | Ref | 1.00 | Ref | 1.00 | Ref |
| 25–34 | | 2.27 | 1.62–3.19 | 2.35 | 1.68–3.30 | 0.99 | 0.75–1.31 | 1.05 | 0.79–1.39 | 1.66 | 1.00–2.77 | 1.77 | 1.06–2.96 |
| 35–44 | | 1.27 | 0.78–2.08 | 1.29 | 0.79–2.12 | 0.48 | 0.29–0.77 | 0.51 | 0.31–0.83 | 1.23 | 0.60–2.55 | 1.30 | 0.63–2.69 |
| **Residence** | | | | | | | | | | | | | |
| Urban | | 1.00 | Ref | 1.00 | Ref | 1.00 | Ref | 1.00 | Ref | 1.00 | Ref | 1.00 | Ref |
| Rural | | 0.63 | 0.40–0.98 | 0.66 | 0.42–1.04 | 0.48 | 0.29–0.77 | 0.49 | 0.29–0.83 | 0.36 | 0.14–0.92 | 0.40 | 0.16–1.03 |
| **Education** | | | | | | | | | | | | | |
| Primary or less | | 1.00 | Ref | 1.00 | Ref | 1.00 | Ref | 1.00 | Ref | 1.00 | Ref | 1.00 | Ref |
| Secondary | | 0.49 | 0.32–0.75 | 0.48 | 0.31–0.74 | 0.77 | 0.46–1.28 | 0.80 | 0.47–1.34 | 1.43 | 0.55–3.75 | 1.48 | 0.56–3.91) |
| Higher | | 0.22 | 0.13–0.36 | 0.20 | 0.12–0.33 | 0.49 | 0.28–0.86 | 0.47 | 0.26–0.82 | 0.73 | 0.26–2.08 | 0.67 | 0.23–1.92 |
| **Wealth index** | | | | | | | | | | | | | |
| Poorest | | 1.00 | Ref | 1.00 | Ref | 1.00 | Ref | 1.00 | Ref | 1.00 | Ref | 1.00 | Ref |
| Poor | | 1.39 | 0.92–2.09 | 1.44 | 0.95–2.19 | 1.25 | 0.77–2.01 | 1.31 | 0.81–2.11 | 0.50 | 0.22–1.13 | 0.54 | 0.24–1.22 |
| Middle | | 1.64 | 1.08–2.48 | 1.74 | 1.14–2.66 | 2.61 | 1.69–4.02 | 2.57 | 1.66–3.99 | 1.71 | 0.93–3.16 | 1.80 | 0.97–3.35 |
| Rich | | 1.61 | 1.03–2.52 | 1.67 | 1.06–2.62 | 2.26 | 1.42–3.59 | 2.26 | 1.42–3.60 | 1.19 | 0.59–2.39 | 1.23 | 0.61–2.48 |
| Richest | | 1.49 | 0.87–2.56 | 1.56 | 0.90–2.69 | 4.20 | 2.61–6.75 | 4.24 | 2.61–6.87 | 1.63 | 0.77–3.46 | 1.73 | 0.80–3.70 |
| **Year** | | | | | | | | | | | | | |
| 2007 | | -- | -- | 1.00 | Ref | -- | -- | 1.00 | Ref | -- | -- | 1.00 | Ref |
| 2009 | | -- | -- | 0.70 | 0.44–1.09 | -- | -- | 1.56 | 0.99–2.46 | -- | -- | 1.09 | 0.47–2.53 |
| 2012 | | -- | -- | 1.17 | 0.78–1.74 | -- | -- | 2.65 | 1.74–4.03 | -- | -- | 2.35 | 1.14–4.87 |
| 2017 | | -- | -- | 1.55 | 1.08–2.21 | -- | -- | 2.59 | 1.72–3.83 | -- | -- | 3.42 | 1.75–6.67 |

*AOR: Adjusted odds ratio

^C.I.: 95% Confidence Interval

seems to have emerged throughout the last decade and is establishing itself as a new method of tobacco use that is rarely reported in the literature. Household wealth and maternal education seem to have a critical role in designing tobacco control interventions and measures. However, social patterning of wealth and education exhibited a complex array in relation to tobacco use. Education clearly protected against cigarette smoking but does not exert the same protective effect on waterpipe use. On the other hand, increasing household wealth exerted a negative effect on cigarette, waterpipe, and dual use.

Early last decade, waterpipe tobacco use re-emerged as a "safe" and socially acceptable tobacco use method [40] that exposed several youth, especially girls, to tobacco dependence and harm [41]. The reported increased prevalence estimates of cigarette smoking among Arab boys and girls [42], combined with the evident waterpipe use early initiation and prolonged maintenance [43], centered waterpipe use as a potential contributor to increasing the prevalence of tobacco use and pointed to an emerging public health problem. These findings may reflect the demonstrated increase in the prevalence estimates of waterpipe and dual smoking reported in the current study. Youth limited social "immunity" against waterpipe use, reported between 2000 and 2010, may have facilitated maintaining the smoking habit at a later age and during pregnancy. Analysis of the Global Youth Tobacco Survey data from the last decade indicated narrowing gaps of tobacco use, especially waterpipe, between boys and girls, and suggested substantial future increases in tobacco use among young women [20, 44, 45]. To be

able to further our understanding of the maintenance of tobacco use during pregnancy, further research should establish a cohort of women and follow them up to see how changes in tobacco use patterns are established over a prolonged period of time and also to focus more on qualitative investigations to shed the light on the connectedness of tobacco products across different age groups.

Our results show that the prevalence estimates of current cigarette smoking among pregnant women in Jordan ranged between 3.0% to 5.7% during the last ten years, a rate that is lower than the 7.9% reported by a study conducted in the north and middle of Jordan around 2012 (28). This difference is probably due to variations in the sample size, location, and sampling technique of both studies. Higher rate was reported in Lebanon with about 17% of pregnant women reporting cigarette smoking in 2004 [27]. This could be due to the fact that Lebanon has the highest rate of young women smoking cigarettes in the Arab region [20]. Worldwide, a meta-analysis reported the highest prevalence estimates of cigarette smoking during pregnancy in different countries, namely Ireland (38.4%), Uruguay (29.7%), Bulgaria (29.4%), Spain (26.0%), and Denmark (25.2%) while the lowest estimates were found in Tanzania (0.2%), Burundi (0.3%), St Lucia (0.3%), Sri Lanka (0.3%), and Malawi (0.3%) [25].

In contrast to cigarette smoking, current waterpipe prevalence estimates have shown a steady increase between 2007 and 2012. Our findings show that the estimates of current waterpipe use surpassed the estimates of current cigarette smoking among pregnant women in Jordan during the last ten years. Notably, it became double the prevalence of current cigarette smoking in 2012. This may indicate that women generally tend to prefer waterpipe use over cigarette smoking, and is supported by a study conducted among university students in Jordan which reported that 53% of female students prefer waterpipe smoking-only use [46]. Interestingly, in Jordan, there is a huge expansion in the number of cafes that offer waterpipe due to its growing popularity among women and teenagers [47]. Its popularity stemmed from the fact that waterpipe use is perceived as less harmful, and more socially acceptable, than cigarette smoking, combined with lack of media campaigns about waterpipe-related health hazards [23]. Water filtration and fruit flavors in the waterpipe introduced it as a *healthy* alternative to cigarette smoking [48]. In addition, youth and women who smoke waterpipe report that it is less addictive than cigarettes [49]. When Jordanian pregnant women were asked about tobacco use addiction, the majority reported cigarettes as being addictive, whereas only 55.1% believed waterpipe to be addictive [28]. The misconception that waterpipe use is a healthy behavior, along with its social acceptability and lack of implemented anti-tobacco policies, were suggested to bridge the gap in waterpipe use between boys and girls and to fuel waterpipe use among girls around 2013 [20, 44, 50, 51].

Our findings showed that cigarette smoking was predominantly present among pregnant women in their later twenties and early thirties which could be due to the fact that pregnant women in Jordan tend to smoke cigarettes later in their life because of the stigmatization and social restrictions placed on younger women. Our results, on the other hand, showed that older maternal age predicted lower risk of waterpipe smoking. This indicates that older age is considered a protective element from waterpipe smoking which is in line with other studies that demonstrated that older maternal age was associated with a higher likelihood of smoking cessation [34, 35, 52]. With respect to education, our results demonstrated a significant inverse association between maternal education and cigarette and waterpipe smoking. This is consistent with the other studies revealing that higher maternal education was protective against cigarette smoking during pregnancy [53–55]. Thus, higher education may serve as a motive for pregnant women not to uptake not only cigarette smoking but also waterpipe use. Interestingly, our results show a positive relationship between wealth index and cigarette and waterpipe smoking among pregnant women. Cigarette smoking mainly increased among pregnant

women living in middle and rich households. This indicates that higher socio-demographic status has a deleterious effect on smoking. This finding is inconsistent with the studies demonstrating that pregnant women with low income level are more prone to continue cigarette smoking [33, 56, 57]. Similarly, our findings showed higher odds of waterpipe use among pregnant women living in rich households. This indicates that waterpipe smoking is predominant among pregnant women with high socio-demographic status. Studies conducted in the USA, Britain, and Syria targeting adults have linked a favorable socio-demographic status to an increased prevalence of waterpipe use [58–60].

Our results revealed a significant increase in the trend of cigarette and waterpipe smoking between 2007 and 2017. A continuous increase in the odds of the dual use was also demonstrated among pregnant women in Jordan during the last ten years reaching up to 3 times in 2017. These findings are in line with the recent evidence indicating that waterpipe use can serve as a gateway to cigarette smoking initiation [61]. Further, the collision of waterpipe and cigarette use among youth in the Arab states was previously reported. Pooled estimates from the Arab states demonstrated a 3.8% prevalence rate of dual tobacco use. These estimates were higher in boys than girls, although this gap was narrower than that of cigarette-only smoking. Dual use was also more prevalent in older than younger youth and varied considerably by country, with rates being high for both boys and girls in Jordan, Lebanon, and the West Bank [44]. Accordingly, the dual tobacco use trends seem to be consistent in different population subgroups in Jordan and further our understanding of the interconnectedness between cigarette and waterpipe epidemics in the region. This stresses the eminent need to develop culturally responsive prevention and cessation strategies not only for youth but also for pregnant women.

There is a strong evidence and robust data showing that smoking during pregnancy leads to adverse perinatal and long-term outcomes and quitting smoking as early as possible during pregnancy can reduce the expected risks placed on both the mother and her infants. Clear and unambiguous messages on the hazards and addiction characteristics of waterpipe, coupled with accessibility and affordability of smoking cessation services, at the primary healthcare level, are then important to reduce the increasing burden of waterpipe and dual use among pregnant women. It is therefore crucial to initiate anti-tobacco interventional programs starting from the health system through Maternal and Child Health (MCH) services. It is well-known that nearly all pregnant women receive about ten prenatal care visits, by primary healthcare professionals, during the course of pregnancy. As a result, an integrative approach involving MCH professionals will be required in order to provide the appropriate support to pregnant women through counseling so that they can quit tobacco use and avoid the aforementioned adverse health consequences. In the US, public health services recommend the five A's model to be implemented in combatting tobacco use and prevention of tobacco dependence [62]. This model could be applied in MCH services to facilitate smoking cessation among pregnant women utilizing available resources.

Undoubtedly, the preconception period is considered the best time to intervene [63]. Thus preconception cessation strategies targeting women in the child-bearing period are needed to be implemented despite the challenges facing the government of Jordan. These include low perception of risk in the public especially among teenagers because everyone around them smoke, the affordability of tobacco products, and lack of policies, especially waterpipe tailored, that promote tobacco use cessation because of the complex chain of measures a legislation must go through in order to be implemented [47]. It is therefore important to plan a multidisciplinary approach involving both smoking cessation services with MCH services to overcome this public health problem.

In Jordan, the different patterning in tobacco use between cigarettes and waterpipe smoking among pregnant women is very tempting for the design of specific cigarette and waterpipe

tailored interventions. Currently, young women with higher socio-demographic status (wealthiest households) use waterpipe at higher rates and this could have useful implications in targeting this group for waterpipe awareness programs. Moreover, the prevalence estimates of tobacco use, especially waterpipe smoking can serve to inform the stakeholders and policy-makers of the magnitude of the public health problem in Jordan. Unless regulations are strictly applied and enforced, the prevalence of tobacco use will be on the rise leaving pregnant women in the face of maternal and child adverse effects.

This study makes an important contribution to the literature by providing nationally representative estimates of cigarette, waterpipe, and dual use among pregnant women in Jordan during the last decade. However, the study has limitations that merit discussion. It should be mentioned that in a conservative country like Jordan, women smoking is not of social conventions, and this may result in under-reporting. This problem of under-reporting can be reduced in future studies by adapting biomarkers measurement from the study population. Still, the validity of self-reports of tobacco use has been demonstrated elsewhere [64]. Assessing ever-married women may limit generalizability. Still, in a conservative society, like Jordan, marriage outside wedlock is punishable by law and puts women at risk of honor killing [65, 66]. This may eliminate the generalizability limitation of the results.

The fact that our data was fully subtracted from the DHS did not allow assessing the levels of electronic smoking, nor the levels of secondhand smoking exposure. However, electronic cigarettes were not common in Jordan until around 2019. Furthermore, our outcome measure have limited information regarding the patterns of tobacco use, such as the history of tobacco smoking initiation and the volume used each day. For future studies, we recommend the addition of biomarkers measurement as a way of verification to get accurate estimates of tobacco use prevalence and the insertion of new tools to measure the prevalence of electronic cigarette use and secondhand smoke exposure.

## Conclusion

In this study, we were able to provide an update of the prevalence of tobacco smoking during pregnancy in a nationally representative sample and to show the trend through four waves of DHS datasets. We also assessed the association of tobacco use with socio- demographic factors. Our results showed that the tobacco use epidemic is expanding its roots among pregnant women in Jordan through not only waterpipe use but also dual cigarette–waterpipe smoking. The seriousness of this expansion varies considerably by socio-demographic factors. Education showed an inverse relationship with cigarette and waterpipe smoking. Household wealth demonstrated a positive association with cigarette and waterpipe smoking. Tobacco use during pregnancy is a global public health problem that seems to be persistent. This study is among the few studies conducted in the region which make its findings valuable for control measures. Tobacco control efforts in Jordan should focus on lowering tobacco use prevalence taking the socio-demographic determinants into consideration.

## Acknowledgments

The authors would like to thank all those who made this data available for analysis especially ICF and DOS.

## Author Contributions

**Conceptualization:** Khalid A. Kheirallah, Maan M. Alolimat.

**Data curation:** Nuha Shugaa Addin.

**Formal analysis:** Khalid A. Kheirallah.

**Methodology:** Nuha Shugaa Addin.

**Supervision:** Khalid A. Kheirallah.

**Validation:** Khalid A. Kheirallah.

**Visualization:** Khalid A. Kheirallah, Maan M. Alolimat.

**Writing – original draft:** Nuha Shugaa Addin.

**Writing – review & editing:** Khalid A. Kheirallah, Maan M. Alolimat.

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
