## [Decision Letter · Decision Letter 0]

21 Apr 2021

PONE-D-21-02978

Trends of maternal waterpipe, cigarettes, and dual tobacco smoking in Jordan. A decade of lost opportunities.

PLOS ONE

Dear Dr. Kheirallah,

Thank you for submitting your manuscript to PLOS ONE. After careful consideration, we feel that it has merit but does not fully meet PLOS ONE’s publication criteria as it currently stands. Therefore, we invite you to submit a revised version of the manuscript that addresses the points raised during the review process.

Your manuscript needs language editing for typing, grammar and punctuation errors. 

We look forward to receiving your revised manuscript.

Kind regards,

Eman Sobh, M.D.

Academic Editor

PLOS ONE

Journal Requirements:

'The funders had no role in study design, data collection and analysis, decision to publish, or preparation of the manuscript'

Reviewers' comments:

Reviewer's Responses to Questions

**Comments to the Author**

1. Is the manuscript technically sound, and do the data support the conclusions?

Reviewer #1: Yes

Reviewer #2: Yes

2. Has the statistical analysis been performed appropriately and rigorously? 

Reviewer #1: Yes

Reviewer #2: Yes

3. Have the authors made all data underlying the findings in their manuscript fully available?

Reviewer #1: No

Reviewer #2: Yes

4. Is the manuscript presented in an intelligible fashion and written in standard English?

Reviewer #1: Yes

Reviewer #2: Yes

5. Review Comments to the Author

Reviewer #1: The study is well written, No further comments of recommendations are requested

You presented your findings in a very good way and findings may help decision makers to find a problem of increased prevalence of Water Pipe (Shisha) Smoking.

Reviewer #2: an excellent subject dealing with smoking pattern among pregnant ladies in your contrary with well illustrated results, discussion and conclusions. The language is very good and statistical analysis is excellent.

6. PLOS authors have the option to publish the peer review history of their article (what does this mean?). If published, this will include your full peer review and any attached files.

Reviewer #1: **Yes: **Tamer Hifnawy

Reviewer #2: **Yes: **Gamal Agmy

---

## [Author Response · Author response to Decision Letter 0]

24 Apr 2021

Dear Editor, 

Thank you for giving us the opportunity to further work on this manuscript while it is being considered for publication in PlosOne. We believe there were no notes from the two reviews and the only thing was an editorial note as below. 

We have updated the manuscript and edited the author list as per the guidelines. Please advise if there were anything specific that needs addressing. 

Reference list has been updated as per the journal guideline. Specially, references 2-5 were updated as there was an issues with them. 

'The funders had no role in study design, data collection and analysis, decision to publish, or preparation of the manuscript'

a. Please clarify the sources of funding (financial or material support) for your study. List the grants or organizations that supported your study, including funding received from your institution.

d. If you did not receive any funding for this study, please state: “The authors received no specific funding for this work.”

Thanks for this update. Please note the below statement. 

The authors received no specific funding for this work.

As for the reviewers comments, the only thing in the email sent to us was a note from reviewer one regarding lack of data. We indicated that DHS data is publically available and downloadable at measuredhs.org. 

Reviewers' comments:

Reviewer's Responses to Questions

Comments to the Author

1. Is the manuscript technically sound, and do the data support the conclusions?

Reviewer #1: Yes

Reviewer #2: Yes

2. Has the statistical analysis been performed appropriately and rigorously?

Reviewer #1: Yes

Reviewer #2: Yes

3. Have the authors made all data underlying the findings in their manuscript fully available?

Reviewer #1: No

We believe the data is publically available from the dhs website. 

Reviewer #2: Yes

4. Is the manuscript presented in an intelligible fashion and written in standard English?

Reviewer #1: Yes

Reviewer #2: Yes

5. Review Comments to the Author

Reviewer #1: The study is well written, No further comments of recommendations are requested

You presented your findings in a very good way and findings may help decision makers to find a problem of increased prevalence of Water Pipe (Shisha) Smoking.

Reviewer #2: an excellent subject dealing with smoking pattern among pregnant ladies in your contrary with well illustrated results, discussion and conclusions. The language is very good and statistical analysis is excellent.

6. PLOS authors have the option to publish the peer review history of their article (what does this mean?). If published, this will include your full peer review and any attached files.

Do you want your identity to be public for this peer review? For information about this choice, including consent withdrawal, please see our Privacy Policy.

Reviewer #1: Yes: Tamer Hifnawy

Reviewer #2: Yes: Gamal Agmy

---

## [Decision Letter · Decision Letter 1]

4 May 2021

PONE-D-21-02978R1

Trends of maternal waterpipe, cigarettes, and dual tobacco smoking in Jordan. A decade of lost opportunities.

PLOS ONE

Dear Dr. Kheirallah,

Thank you for submitting your manuscript to PLOS ONE. After careful consideration, we feel that it has merit but does not fully meet PLOS ONE’s publication criteria as it currently stands. Therefore, we invite you to submit a revised version of the manuscript that addresses the points raised during the review process.

Please respond to comments from statistical reviewer. Your submission still requires substantial editing for English grammar and usage. We ask that you please have the manuscript copyedited by either a native-English speaking colleague or a professional copy-editing service. While you may approach any qualified individual or any professional scientific editing service of your choice, PLOS has partnered with American Journal Experts (AJE) to provide discounted services to PLOS authors. AJE has extensive experience helping authors meet PLOS guidelines and can provide language editing, translation, manuscript formatting, and figure formatting to ensure your manuscript meets our submission guidelines. If the PLOS editorial team finds any language issues in text that AJE has edited, AJE will re-edit the text for free. To take advantage of this special partnership, use the following link: https://www.aje.com/go/plos/

In the revised file we noted you have added a new author; however there is no substantial efforts has been made in the revised version. please refer to authorship criteria and those who did not fulfill authorship criteria can be added in acknowledgment section.

We look forward to receiving your revised manuscript.

Kind regards,

Eman Sobh, M.D.

Academic Editor

PLOS ONE

Journal Requirements:

Reviewers' comments:

Reviewer's Responses to Questions

**Comments to the Author**

1. If the authors have adequately addressed your comments raised in a previous round of review and you feel that this manuscript is now acceptable for publication, you may indicate that here to bypass the “Comments to the Author” section, enter your conflict of interest statement in the “Confidential to Editor” section, and submit your "Accept" recommendation.

Reviewer #3: (No Response)

2. Is the manuscript technically sound, and do the data support the conclusions?

Reviewer #3: Yes

3. Has the statistical analysis been performed appropriately and rigorously? 

Reviewer #3: Yes

4. Have the authors made all data underlying the findings in their manuscript fully available?

Reviewer #3: Yes

5. Is the manuscript presented in an intelligible fashion and written in standard English?

Reviewer #3: Yes

6. Review Comments to the Author

Reviewer #3: I will focus on methods and reporting. Please note that this is the first time I review this paper.

Overall, clear and well written paper, well done.

Major

1) There is no mention of missing data. Were the data 100% complete? if yes then state so. if not, why were multiple imputation models considered?

2) Explain the need for the two modelling approaches. why do you need a time adjusted and an unadjusted model? in other words, what does the unadjusted model add, since the time adjusted model is essential in quantifying any increases over time in the outcomes of interest? and to rephrase again, why don't you drop the time unadjusted model altogether, why is it essential?

Minor

1) I struggled to follow the results section in the abstract. it seemed like other outcomes (not previously mentioned) were reported, e.g. "current tobacco use". Please use consistent terminology between the methods and results sections, even if repetitive. also some of the statements are not exactly consistent with the numbers reported e.g. stable when there is an increase and an increase when there is a very large increase.

2) discuss the married only status for the respondents as minor or major limitation. In the UK it would be a major limitation since pregnancy is not uncommon outside wedlock. Please explain that this is extremely unlikely in Jordan, as I suspect it is, and possibly reference some evidence to that.

3) In the methods section clarify if time was included in the model as a continuous or categorical covariate. Also there is a mention of interaction between socio-economic variables and time. Need clarity whether an interaction term was included (in which case the interpretation of the main effects becomes very difficult if not impossible), or if an interaction was not included (in which case the authors should remove the mention of interaction, since they don't mean statistical interaction, and the term is confusing.

4) Provide a reference for the VIF threshold. I've seen 4 used, and 10 as a more relaxed threshold, but never seen 5 used.

5) Can't have a p-value of 0. when small report as <0.001

6) restructure tables to include the 95% CI with the estimate. e.g. 1.00 (0.80 to 1.20). I would drop the p-values altogether since they are superfluous.

7) The paper is lacking in visual evidence, the authors should consider appropriate graphs to better communicate their findings.

7. PLOS authors have the option to publish the peer review history of their article (what does this mean?). If published, this will include your full peer review and any attached files.

Reviewer #3: No

---

## [Author Response · Author response to Decision Letter 1]

6 May 2021

Please see Response to Reviewers letter at the end of this PDF.

---

## [Decision Letter · Decision Letter 2]

10 Jun 2021

Trends of maternal waterpipe, cigarettes, and dual tobacco smoking in Jordan. A decade of lost opportunities.

PONE-D-21-02978R2

Dear Dr. Kheirallah,

We’re pleased to inform you that your manuscript has been judged scientifically suitable for publication and will be formally accepted for publication once it meets all outstanding technical requirements.

Kind regards,

Eman Sobh, M.D.

Academic Editor

PLOS ONE

Additional Editor Comments (optional):

Reviewers' comments:

Reviewer's Responses to Questions

**Comments to the Author**

1. If the authors have adequately addressed your comments raised in a previous round of review and you feel that this manuscript is now acceptable for publication, you may indicate that here to bypass the “Comments to the Author” section, enter your conflict of interest statement in the “Confidential to Editor” section, and submit your "Accept" recommendation.

Reviewer #3: All comments have been addressed

2. Is the manuscript technically sound, and do the data support the conclusions?

Reviewer #3: Yes

3. Has the statistical analysis been performed appropriately and rigorously? 

Reviewer #3: Yes

4. Have the authors made all data underlying the findings in their manuscript fully available?

Reviewer #3: Yes

5. Is the manuscript presented in an intelligible fashion and written in standard English?

Reviewer #3: Yes

6. Review Comments to the Author

Reviewer #3: I have no further comments. the authors' answers are convincing and clear and the resulting changes to the paper are satisfactory

7. PLOS authors have the option to publish the peer review history of their article (what does this mean?). If published, this will include your full peer review and any attached files.

Reviewer #3: No

---

## [Editor Report · Acceptance letter]

1 Jul 2021

PONE-D-21-02978R2 

Trends of maternal waterpipe, cigarettes, and dual tobacco smoking in Jordan. A decade of lost opportunities. 

Dear Dr. Kheirallah:

I'm pleased to inform you that your manuscript has been deemed suitable for publication in PLOS ONE. Congratulations! Your manuscript is now with our production department. 

Kind regards, 

on behalf of

Dr. Eman Sobh 

Academic Editor

PLOS ONE